# The Measurement of Contact Angle, pH, and Conductivity of Artificial Saliva and Mouthwashes on Enamel, Glass-Ionomer, and Composite Dental Materials

**DOI:** 10.3390/ma15134533

**Published:** 2022-06-28

**Authors:** Saima Qureshi, Lazar Milić, Bojan Petrović, Marija Vejin, Sanja Kojić, Stefan Jarić, Goran Stojanović

**Affiliations:** 1Department of Power, Electronics and Telecommunications, Faculty of Technical Sciences, University of Novi Sad, Fruškogorska, 11, 21000 Novi Sad, Serbia; lmilic@uns.ac.rs (L.M.); vejinmarija@uns.ac.rs (M.V.); sanjakojic@uns.ac.rs (S.K.); sgoran@uns.ac.rs (G.S.); 2Department of Dentistry, Faculty of Medicine, University of Novi Sad, Hajduk Veljkova, 3, 21000 Novi Sad, Serbia; 3Institute BioSense, University of Novi Sad, Dr. Zorana Đinđića, 1, 21000 Novi Sad, Serbia; sjaric@biosense.rs

**Keywords:** contact angle, conductivity, MATLAB, saliva, wetting, enamel, glass ionomer, composite

## Abstract

The aim of this study was to tackle the topic of appropriate recommendations for artificial-saliva and mouthwash usage. The contact angle, pH, and conductivity of two artificial saliva solutions, four mouthwashes, and their mixtures on enamel, glass-ionomer, and composite dental materials were measured. The measurements were conducted with a MATLAB algorithm to minimize human error. The obtained values for the contact angle were in the range from 7.98° to 52.6°, and they showed completely nonlinear and nonuniform behavior for all investigated liquids and on all investigated substrates. Results reveal statistically significant differences among all tested liquids on all investigated substrates (*p* < 0.05). pH values ranged from 1.54 to 7.01. A wide range of conductivity values were observed, from 1205µS/cm in the saliva-stimulating solution to 6679 mS/cm in the artificial saliva. Spearman’s test showed a moderate positive correlation between the pH and conductivity of the tested fluids (R = 0.7108). A comparison of the data obtained using Image J software and the MATLAB algorithm showed consistency, not exceeding 5% error. When an experiment uses human material and bioactive materials THAT are used in biomedicine as substrates, an additional definition of protocols is highly recommended for future research on this topic.

## 1. Introduction

The angle formed tangential to the liquid surface at the line where three phases meet and the plane of the surface of solid material on which liquid substance resides or moves is the contact angle. The contact angle is usually measured by sessile drop instruments by curve fitting the whole drop profile, employing the Young–Laplace equation [1,2]. A lower contact angle indicates increased wettability or higher hydrophilicity, while a higher contact angle indicates limited wettability or higher hydrophobicity.

Humans secrete between 1 and 1.5 L of saliva per day. Water (99.0–99.5%), and inorganic and organic compounds such as immunoglobulins, proteins, enzymes, mucins, and nitrogenous compounds (0.5–1%) comprise the majority of saliva [3]. The interaction between saliva and morphological structures, mucosa, and hard dental tissues is critical for overall oral health and physiology. Understanding how saliva interacts with materials can help us in better understanding how enamel and restorative materials react on the surface [4,5].

Substrates with different surface characteristics are present in the oral cavity (enamel, restorative materials, implants, orthodontic and prosthodontic appliances). The wettability of the oral substrate should include the measurement of the contact angle of saliva, artificial saliva, and various mouthwash solutions on enamel, glass ionomer cement, composite material, and other materials. The interpretation of wettability can give valuable information about the interaction between surfaces of the oral cavity and liquids that are widely used [6,7]. In addition, a large contact angle indicates that the adhesion between liquid and solid is weak, which is extremely important when it comes to the effect of bacterial adhesion on the enamel surface, but also for the effective delivery of active ingredients of mouthwashes that should adhere on the enamel surface and exhibit an antibacterial or remineralizing effect [8,9].

Some studies have found that bacterial adhesion is higher on hydrophobic surfaces, while others have found that highly hydrophilic substrates promote biofilm development [8]. According to Sang et al. [9], new dental materials should be designed to tune the thickness, composition, and structure of the adsorbed salivary pellicle to control bacterial attachment.

Artificial saliva and mouthwashes are primarily designed for persons with hyposalivation, i.e., dry mouth, in order to alleviate symptoms experienced during eating, swallowing, and talking.

In applications where a biomaterial surface comes into contact with a liquid phase, such as restorative materials commonly used for fillings, it is critical to understand not only the surface characteristics under normal experimental conditions, but also the effects of exposure to the liquid medium. As a result, tested materials should be exposed to a variety of medium of interest in order to determine a surface characteristic under conditions that are as close to the real world as possible. The behavior of dental materials in the oral environment serves as a foundation for determining whether or not they should be used.

The insufficient wetting of the mucosa by water is clinically significant and corresponds to the experience of patients who report that water does not adequately moisten their mouths. It is preferable to use a mucin-containing saliva substitute. Clinical trials and a subsequent comprehensive review of carboxyl-methylcellulose and mucin-containing saliva substitutes [10] also confirmed this. The rheological and remineralizing properties of saliva substitutes are two other properties that may influence their effectiveness in reducing xerostomia complaints in patients. Because both carboxy-methylcellulose and mucin-containing saliva substitutes have potential rehardening properties, they may play a role in xerostomia-related dental caries control. Mucin-containing saliva substitutes should have rheological properties comparable to whole human saliva. Wettability determines the functional group rearrangement at the surface of biomaterials in contact with cells [11,12,13]. Bacterial adhesion occurs more frequently on hydrophilic surfaces with high surface free energy values than it does on hydrophobic surfaces [11,12,13]. Although an ideal saliva substitute has similar rheological and biochemical properties to natural human saliva [10], adding antimicrobials to a solution with rheological properties similar to human saliva may be a better solution, and this approach is currently feasible. Home remedies such as water, olive oil, and milk, and saliva substitutes used as mouthwashes, rinses, gels, and sprays are examples of palliative oral care products [14,15,16]. They are used to relieve oral discomfort by moistening the oral mucosa without stimulating salivary flow. Another important issue that has been extensively discussed for saliva substitutes and saliva stimulants used with various antibacterial mouthwashes is their effect on enamel and dentin demineralization prevention because they can significantly lower intraoral pH regardless of antibacterial effect. Adjustments must be made while maintaining a balance of osmolality, conductivity, and viscosity, which is extremely difficult to achieve. Saliva is a hypotonic fluid, which is important because it preserves taste and prevents epithelium dehydration. Overall, these problems suggest that there is currently no saliva substitute, saliva stimulant, or mouthwash that adequately mimics the complex properties of natural saliva. As a result, extensive research into this topic is urgently required. The aim of this study was to tackle the topic of appropriate recommendation for artificial-saliva and mouthwash usage. Since, these substances are used in everyday clinical practice, it is important to investigate how they behave on enamel, glass-ionomer, and composites through contact angle, pH, and conductivity analyses. To improve a relatively simple experimental setup and the quality of measurements, the MATLAB algorithm for measurement was used instead of routine procedures and commercially available software.

## 2. Materials and Methods

Three donated intact human teeth had their labial and buccal surfaces cut with a diamond disc, washed with abrasive paper of fineness ranging between 800 and 1200, and polished with Al_2_O_3_ powder in distilled water.

After that, demineralized water was used to wash the polished surface. Artificial saliva (AS1) composed of carboxymethyl cellulose and produced according to the recipe of the Pharmacy Institution Belgrade (registered under the Republic of Serbia’s master preparations) was investigated in this study. Separately, a glycerin/citric acid solution was produced. According to the recipe, 388 g of glycerin and 25 g of citric acid were used to produce 1 kilogram of the solution (registered under master preparations of the Republic of Serbia) (AS2) with four different mouthwashes:(1)0.1% chlorhexidine solution (HX) (Eludril Classic, Pierre Fabre Medicament, Bologna, France);(2)Elmex, mouthwash (F) (Gaba International, Therwill, Basel Stadt, Switzerland ) with the active ingredients of 100 ppm amine fluoride and 150 ppm sodium fluoride;(3)Listerine (L) (Johnson & Johnson, New Brunswick, NJ, USA, Cool Mint Listerine);(4)Dentadent (DD) (a nonalcohol formulation containing aluminum lactate, sodium fluoride, and chlorhexidine (Lilly Drogerie, Belgrade, Serbia).

A total of 14 different solutions were tested on the surface of intact enamel, glass-ionomer, and composite material in this study (artificial saliva solution, saliva stimulant solution, four mouthwash solutions, and eight solutions (1:1) of each mouthwash solution with the artificial saliva and saliva stimulant).

The tested glass ionomer material (FIX, Fuji IX GP conventional glass ionomer restorative cement, GC Int, Tokyo, Japan) was prepared according to the manufacturer’s recommendations, similar to what was reported [17]. Capsules were activated immediately before mixing and placed into the amalgamator device for 10 s of mixing. Afterwards, the activated capsules were placed into the original capsule applier. The material was placed into Teflon molds immediately after mixing. The Teflon molds were cylindrical, with a diameter of 15 mm and a thickness of 10 mm. During the setting of the experimental disks, the bottom and top of each mold was roofed using glass plates and hand pressure for 10 min. Immediately upon the complete setting of the materials, the disks were taken out from the molds and polished using Sof-Lex discs 8691-F (3M ESPE AG, Seefeld, Germany).

The tested composite material, Gradia Direct (GC America, Alsip, IL, USA), was prepared as follows. The material was placed into the same Teflon molds used for glass ionomer cement and polymerized for 40 s with a light-curing unit operating in standard mode and emitting no less than 600 mW/cm^2^. Immediately upon the complete setting of the materials, the disks were taken from the molds and polished using Sof-Lex discs 8691-F (3M ESPE AG, Seefeld, Germany).

The pH and conductivity values of the solutions were obtained from pH and conductivity measurements performed with a commercial pH meter and conductometer system (pH/ISE/EC Meter, Hanna Instruments, HI5222, Smithfield, RI, USA).

A standard reference glass electrode with a 3.5 M KCl + AgCl electrolyte was used to measure pH of solutions. Prior to the measurements, the pH reference electrode was calibrated according to the manufacturer’s propositions by two standard electrolytes with pH 7.01 and 4.01 (with pH 0.01 accuracy). After the calibration, the measurements were performed with a simple electrode immersion into the investigated solution; between each measurement, the electrode was rinsed with DI water and carefully dried to avoid false signals due to electrostatics.

On the other hand, conductivity values were obtained with a four-ring conductivity probe electrode. As with the pH electrode, this probe was calibrated with standard electrolytes of 84 and 1413 μS/cm conductivities, and measurements were performed with the proper immersion of the probe into the solution. Between each measurement, the probe was thoroughly rinsed with DI water and dried for the next measurement.

The contact angle was measured in accordance with our previous reports (Figure 1) [18,19]. All static contact angle experiments were performed at room temperature, 25 °C, using a customized setup. The experimental setup consisted of a camera connected to a USB port on a laptop on one side and fixed in place on the other (Figure 2). Movement errors were minimized using an antivibration plate on which the camera and plate for measuring were fixed. With the specimen fixed on the plate, the dripping could commence. Behind the specimen, a light was installed, and three different specimens were used. Aside from that, 14 different liquids were mixed. Each experiment was repeated 10 times for each combination of specimen and liquid. The taken specimen volume was 3 µL and it was dropped vertically on the substrate from 2 cm height. Ten images for contact angle measurement were captured after 10 s. All images were taken by dropping the liquid at different locations of the specimen with different illumination, shading, and focus levels. Final calculations were conducted by taking the average of the measured contact angles. Each combination had its respective folder where 30 images were saved in total. After that, the calculation process was concluded. The contact angle measurements were performed using Image J software using an automated MATLAB algorithm designed specifically for the purposes of the present investigation.

The contact angle calculation algorithm was divided into four sections, each of which was integrated into a MATLAB GUI where the folder path containing the images for contact angle measurement was entered. Following the preceding step, the algorithm was applied to each image. The algorithm’s four main components were as follows: (a) image preparation and edge extraction; (b) line detection using Hough transform; (c) circular Hough transform; (d) calculation of contact angle. The first section was divided into five subsections: (a) loading the image; (b) converting the image into grayscale and doubling its values; (c) forming the Sobel filter and calculating the first derivative of the image; (d) defining a threshold; (e) using morphological transformations to thin out the resulting edges (Figure 3) [18,19]. Even though the Canny edge detector is predominantly used in feature definition, in this algorithm, the Sobel filter was applied in conjunction with a threshold. The dominant reason is that the Sobel filter showed greater consistency in edge extraction over a wide range of contact angle images, with no problem in eliminating redundant fine details, which formed closed loops when the Canny edge detector was applied.

With the goal of eliminating thick lines that were the result of applying the previously mentioned edge extraction filter and threshold on an image with blurred lines, the process of thinning the edges came afterwards using morphological transformations.

The next two segments, line and circle extraction, were conducted using the Hough transform. For line extraction, Equation (1) was used. The extracted line was then drawn as a visual representation of the border between substrates.
(1)y=kx+n

The circle extraction was more complex, as it was designed to search for the best radius and the best coordinates using Equations (2)–(5) and the algorithm described in [18]. This was critical in completely eliminating any user interaction with the image.
(2)$${\left(x-p\right)}^{2}+{\left(y-q\right)}^{2}=R^{2}$$
(3)x=p+Rsin(θ)
(4)y=q+Rcos(θ)
(5)θ∈[0, 360°]

Lastly, the intersecting points of these two features were found, and two tangents were drawn on the circular feature. The angle between the horizontal line, which represents the border between the solid and droplet, and the drawn tangents was the contact angle. Inner contact angles were chosen for this analysis.

The transformation of the edge image using the Hough transform for the purpose of line detection is shown in Figure 3c. For the purpose of avoiding undefined regions in the k–n parameter space, the transform was implemented in the polar coordinate ρ–ϑ system; Equations (6)–(8) were used for calculating k and n as described in [18]. Afterwards, the extracted line was drawn as a visual representation of the border between substrates.
(6)xcos(θ)+ysin(θ)=ρ
(7)k=−cot(θ)
(8)n=ρsin(θ)

Figure 3d shows the edge image in the p–q space, mapped using Equation (2). The pixel (or a group of joint pixels) with the highest intensity in the p–q space represents the coordinates of the circle center in the x–y space for a given radius. This was critical in completely eliminating any user interaction with the image.

After the algorithm finishes extracting the contact angle from one image, it is written in an Excel file. The columns of the mentioned file represented the left and right contact angles, while each row represented each image in the designated folder. The process was repeated for all images in the specified folder.

All contact angle measurements were performed on raw images (Figure 4) using both Image J software and the MATLAB algorithm, and the obtained data were compared. In summary of the data obtained in this experimental study, nominal and categorical variables are presented as number and percentage, while continuous variables are presented as mean with standard deviation. The chi-squared, Kruskal–Wallis, and one-way ANOVA with Tukey HSD tests were used to compare the two groups. For statistical analysis, open-source statistical program Jamovi Project (2021), Jamovi (Version 0.9.2.8), retrieved from https://www.jamovi.org (accessed on 15 December 2021), was used with the significance level set to be 0.05.

## 3. Results

Figure 5 and Table 1 depict average values and standard deviations of all 14 liquids on all 3 investigated substrates. The contact angle values ranged from 13.8 (observed between Listerine and glass-ionomer cement) and 52.63 (recorded between artificial saliva and enamel).

Table 2 shows the pH and conductivity values of 14 tested solutions at room temperature. The pH values ranged from 1.54, which was observed with salivation stimulants, to 7.01, with the artificial-saliva and Dentodent solutions. A wide range of conductivity values were obtained, from 1205 mS/cm in saliva stimulators to 6679 mS/cm in artificial saliva.

Figure 6 shows the completely nonlinear and nonuniform behavior of the contact angles for all investigated liquids. AS1 and solutions with AS1 had a larger contact angle than that of AS2 and its solutions.

As shown in Table 3, data distribution had significantly deviated from normal distribution, so nonparametric statistical analyses were carried out. Regarding enamel, the lowest contact angle was recorded with chlorhexidine solution (13.7), and the largest with artificial saliva (52.6). In contrast to that, the range of the contact angle in the composite material was between 16.4 (with Listerine) and 39.8 (with the mixture of artificial saliva and Dentodent solution). Lastly, the values recorded with the glass-ionomer specimens were in the range from 7.98 (with artificial saliva and Listerine) to 45.9 (with artificial saliva and fluoride solution). Kruskal–Wallis analysis was performed, and the results are shown in Table 3, suggesting that all these differences in mean contact angle values were statistically significant (*p* < 0.05, Table 4).

Figure 7 depicts the relationship among contact angle range, pH range, and conductivity range in 14 analyzed specimens, showing a nonuniform and nonlinear relationship between the investigated parameters, namely, contact angle, pH, and conductivity. AS1 is conductive in its pure form and in solutions, and mainly corresponds to alkaline pH values. The wettability of AS1 was slightly less than the wettability of AS2. Unlike AS1, AS2 had low conductivity, i.e., a low amount of electrolyte that correlated with acidic pH. This acidity was minimally neutralized by diluting with the investigated solution. Multiple statistical methods were employed, and only correlation analysis, namely, Spearman’s test, showed a moderate positive correlation between the pH and conductivity of the tested fluids (R = 0.7108).

Comparison of data obtained using Image J software and the MATLAB algorithm showed consistency, not exceeding 5% error.

## 4. Discussion

In this study, the main goal was to analyze the values of the contact angles of the solutions used daily for oral hygiene and dental care in relation to the three substrates that are most often present in the oral cavity, namely, human enamel and two types of restorative materials, glass-ionomer and composite. Additionally, we tried to establish the relationship between contact angle, and pH and conductivity because the latter two are very important parameters in achieving all preventive and prophylactic properties of the tested agents. In real saliva, pH, conductivity, and contact angle are physiologically balanced in order to preserve oral homeostasis. If necessary, the pH of the saliva increases thanks to its buffering capacity, the contact angle changes with the amount of protein, and the conductivity increases with the increase in electrolytes. The aim of the study was to examine the extent to which saliva substitutes and antiseptic solutions meet these basic physical defensive properties of real saliva. Lastly, an algorithm was developed to automate contact angle measurements.

Contact angle measurement between biological substrates and complex bioactive liquids is not an easy and straightforward task. We considered the fact that contact angle measurements may be affected by numerous experimental conditions, such as drop volume, drop size, surface of the substrate, temperature, and surface impurities. The used drop size Ω affected the contact angle value [20]. In the present investigation, three completely different substrates were analyzed: human dental enamel (consisting of over 95 wt % carbonated apatite), glass-ionomer (based on the reaction of silicate glass powder (calcium-alumino-fluorosilicate glass) and polyacrylic acid, an ionomer), and composite restorative material (consisting of a resin-based oligomer matrix, bisphenol A-glycidyl methacrylate (BISGMA), and an inorganic filler). All three analyzed surfaces could exhibit some impurities, such as heterogeneity and roughness, which could impact the point of contact and cause variations in the contact angle measurements [21]. We overcame the problems of the heterogeneity and complexity of the substrate and liquids in the conducted research by repeating the measurements in identical conditions and comparing the obtained results without extrapolation to the research that used standard substrates or standard solutions with known surface properties. Despite the variances induced by drop volume and surface contaminants, the contact angle approach achieved an accuracy of around 95% [15]. Surface temperatures below 5 °C influence contact angle measurements [22,23,24,25]. All measurements in the present investigation were performed at stable room temperature.

Although an ideal saliva substitute should have rheological and biochemical properties that are similar to natural human saliva [26,27], adding antimicrobials to a solution, or using antiseptic or remineralizing mouthwashes with rheological properties that are similar to human saliva may be a better solution, and this approach is currently feasible. Thus, intraoral surfaces such as enamel and restorative materials within the oral cavity are likely to have substantivity toward various active components, such as fluoride, chlorhexidine, and other chemical plaque control agents [28,29,30].

Understanding the interaction of materials with the saliva can help with the surface treatment parameters for restorative material choice [31]. Abdelasam and his colleagues found contact angles ranging from 13.4 to 48.9 degrees in their investigation, which is completely consistent with our findings [31]. The previously cited review focused on ceramic materials for prosthetic restorations, whereas our study focused on composite and glass ionomer restorative materials. According to the findings, resin nanoceramics is the most hydrophobic dental crown material with a contact angle of 60.5, while zirconia is the most hydrophilic with a contact angle of 20.4. According to Perira and his colleagues, the contact angles of composite materials with water, 1-bromophthalene, and formamine were in the range from 20.90 to 69 [32]. In this regard, the bioadhesion capabilities of dental restorative materials are crucial for their long-term survival in the oral cavity. Oral biofilm production, on the other hand, begins with bacterial attachment to the acquired pellicle, which covers all oral surfaces, including dental restorations, and hides the physicochemical surface properties of dental materials to a degree [33]. Regardless, in situ or in vivo investigations should account for substantial intraindividual and interindividual variability, and many modifying factors such as salivary flow, diet, and oral microorganisms.

Contact angle tests and surface free energy calculations revealed differences in the energy states of dental materials and tissues, and the effect of contact with various liquids on the investigated values. In biological fluid, organic films and cells deposit on the surface of materials, and the primary concern with dental materials is the formation of dental plaque. This holds true for both natural structures and materials used for restorative or prosthetic purposes. Farinone and his coworkers [10] observed that the contact angles of artificial saliva and whole human saliva were comparable in a study on the wetting properties and other physical and chemical features of human saliva and saliva substitutes available on the market. Water, on the other hand, had a substantially larger contact angle with human mucosa than that with full human saliva. Furthermore, water and all types of artificial saliva had much lower contact angles on ground polished enamel than those of complete human saliva. This finding can be partially confirmed by the results from the present investigation.

Firm contact between the adhesive and the adhering surface is required for excellent dental adherence and for the wettability of the substrate. A hydrophobic material does not generally give the best contact with a hydrophilic surface. To improve adhesion to the enamel surface, surface modification for wettability enhancement is required. After various treatments, the water contact angle of the enamel surface changes. There are reports in the literature that each approach alters the wettability of the enamel surface, with the acid-etching pretreatment achieving the highest adhesion efficiency [34,35,36]. We used polished but unconditioned enamel in our research. During everyday clinical procedures, both glass ionomer and composite materials require conditioning with polyacrylic acid or orthophosphoric acid, which can certainly affect the value of the contact angle of the adjacent enamel.

Aside from comparing the surface hydrophilicity and hydrophobicity of the materials, specific emphasis should be on estimating the degree of interaction between material exposed to a simulated biological environment. It is crucial to understand not only the surface properties under typical experimental settings, but also the consequences of exposure to the liquid medium in applications where a biomaterial surface would come into contact with a liquid phase, such as artificial saliva, and various mouthwashes and fillings present in the oral cavity. As a result, the tested materials were subjected to a variety of media in order to determine a surface characteristic that was as close to real-world settings as possible. The ability to determine whether or not some oral mouthwashes should be used in particular situations is based on their behavior in the mouth environment.

Wettability is the ability of a biomaterial to rearrange the functional groups on its surface when it comes into contact with not only the liquid, but also with a cell. Bacterial adherence occurs more frequently on hydrophilic surfaces with high surface free energy values than it does on hydrophobic surfaces. Water contact angles from 40 to 70 degrees have been identified as the most favorable for cell adhesion in polymers. However, defining the range of contact angles for which cells efficiently attach to a material surface is difficult. The adhesion of cells to the surface of a material is a complicated process that is influenced by a number of parameters, such as cell type, surface wettability, roughness, topography, and chemistry.

The contact angle obtained in this study for the composite is very similar to that obtained by Namen et al. [37]. The findings revealed that the adhesive capacities of the materials used as restoratives vary. The surface energy and wettability of various dental materials treated under various storage conditions were used to compare the surface energy and wettability of the initial materials’ surfaces. Water was frequently used as an environment in other studies, which adds to the value of the analysis. The wetting of a surface by a liquid and the ultimate amount of that liquid spreading are important aspects of practical surface chemistry, and there is still much to learn about liquid movement mechanisms. When it comes to mouthwashes, there are reports in the literature about their contact angles with oral structures from both in vivo and in vitro studies. In vivo contact angles ranged from 37° to 54° [38,39]. This finding could be completely confirmed by the results from the present investigation. Additionally, previous reports underline the acidity of some mouthwashes, with the most acidic values reported to be 3.6 and 3.7. In the present investigation, some of the mouthwashes and the widely used saliva stimulant were also extremely acidic, with pH as low as 1.54. To be effective on smooth surfaces, a mouthrinse should cover the whole surface of the tooth, enlarging the contact area and producing a zero contact angle. To be effective in interproximal areas and gingival pockets, a mouthrinse must be able to penetrate deeply. Viscosity, surface tension, and contact angle on the capillary determine a liquid’s ability to permeate pores and capillaries.

Antibacterial, remineralization, rheological, deodorant, and taste criteria are frequently balanced in product composition. These criteria might sometimes clash resulting in unfavorably acidic products. It is difficult to pinpoint why some manufacturers develop mouthrinse formulas that are relatively acidic.

One reason could be because a more acidic product meets the taste requirements more easily or that acidic items are more stable. At a low pH, fluoride ions are more stable because a low pH (less than 5) causes CaF_2_ to lose its stability and dissolve slowly. Higher pH values enable the formation of a protective CaF_2_ coating on enamel from which F-ions are slowly released [40]. The combination of a low pH and a high buffer capacity, on the other hand, appears to be unfavorable and is likely to exacerbate enamel demineralization. In conclusion, the surface tension, in vivo contact angle, viscosity, penetration coefficient, acidity, and buffer capacity of commercially available mouthrinses all differ significantly [38,39]. According to this study and the existing literature, the subgingival activity of mouthrinses can be stimulated by enhancing penetration.

Surface temperatures below 5 and above 120 °C impact contact angle measurements, and experiments showed that, for every 1 °C change in temperature, the contact angle changes by 0.18° [41]. The majority of the similar studies were conducted at room temperature. Furthermore, the temperature in the oral cavity fluctuates constantly between 0 and 55 degrees Celsius depending on the foods and beverages. The effect of temperature on contact angle was ignored in this study because minor temperature changes cause insignificant contact angle changes, and the oral cavity does not have a consistent temperature. For consistency, all materials used in this experiment were kept at room temperature.

The MATLAB algorithm requires very little user interaction with the exception of running the software and image selection. The boundaries on the image should be marked before running the contact angle measurement plugin for ImageJ. The user must mark the boundary with 7 or 5 points depending on the used approximation method. This increases result variability, which is the next parameter. When it comes to determining reliable results, variability is crucial. The MATLAB algorithm is much slower than the ImageJ plugin, taking around 5 min versus 10 s to calculate the contact angle. The only reason for the delay is that fitting the best radius during circular feature extraction takes a long time. [18,19]. This can be changed by changing the end points of the interval that the algorithm traverses while fitting the radius within the algorithm. As the range grows larger, the algorithm becomes slower. Because the main goal of this algorithm was to create a user-friendly way to calculate the contact angle with as little user interaction as possible, the algorithm speed had to be sacrificed. In future iterations of the algorithm, this flaw will be addressed.

## 5. Conclusions

In this paper, we presented the contact angle, pH, and conductivity of two artificial saliva, four mouthwashes, and their mixtures. The liquids were tested on human enamel, glass-ionomer, and composites. The main findings indicate that all tested liquids exhibited hydrophilic behavior, but this behavior significantly varied between the tested liquids, their mixtures, and in regard to the analyzed substrate. Extreme heterogeneity was observed among contact angle, acidity, and conductivity. The observed pH values ranged from 1.54 in the salivation stimulants to 7.01 in the artificial-saliva and chlorhexidine solutions. A wide range of conductivity values were obtained, from 1205 in the saliva stimulators to 6679 mS/cm in the artificial saliva. Despite the fact that the obtained data corelated with the literature, acidity sometimes decreased below the critical levels of pH 5.5, which is clinically relevant. Each liquid exhibited specific behavior on each substrate. The used algorithm aims to remove bias and human error since it is fully automatized, and enables importing a folder of images instead of inserting images one by one.

When an experiment uses human material as substrate, and bioactive materials are used in biomedicine, an additional definition of protocols is highly recommended for the future research of this topic.

## Figures and Tables

**Figure 1 materials-15-04533-f001:**
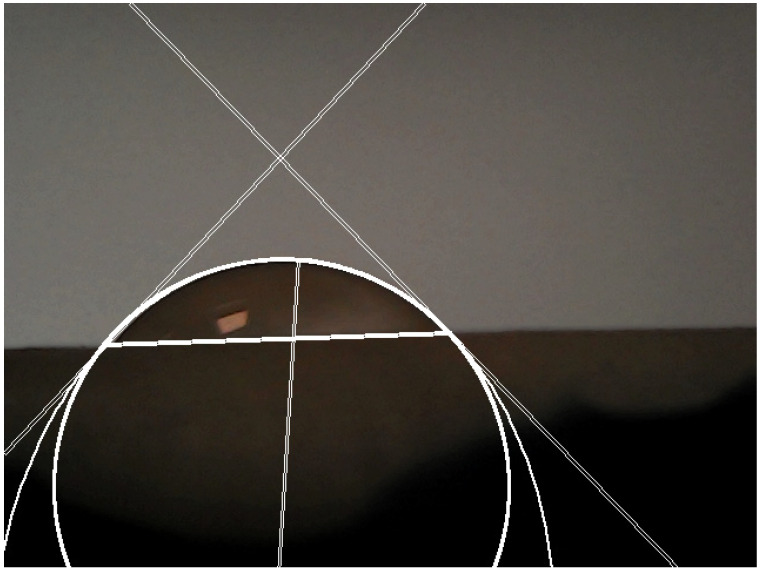
Contact angle measurement using Image J software.

**Figure 2 materials-15-04533-f002:**
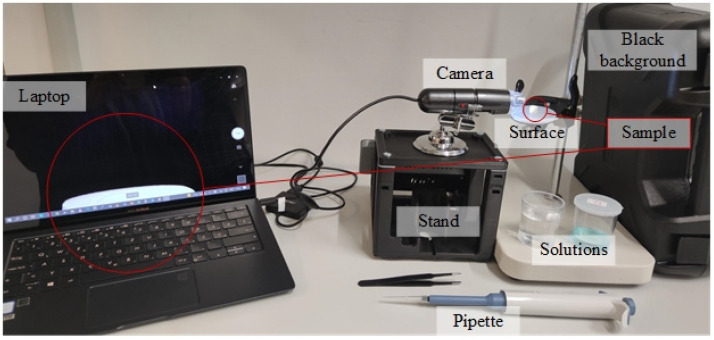
Experimental setup.

**Figure 3 materials-15-04533-f003:**
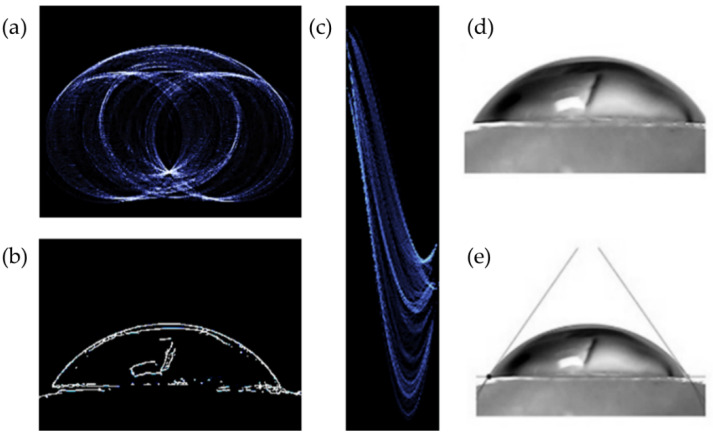
(**a**) Image of the Hough transform for circle detection applied on edge image; (**b**) edge image; (**c**) image of the Hough transform for line detection applied on edge image; (**d**) input image (**e**) output image with angles drawn [13].

**Figure 4 materials-15-04533-f004:**
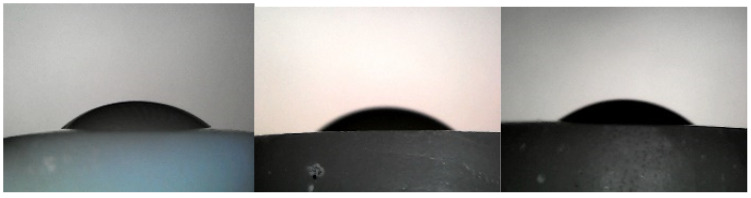
Droplets on the (**left**) enamel surface, (**middle**) glass-ionomer, and (**right**) composite material.

**Figure 5 materials-15-04533-f005:**
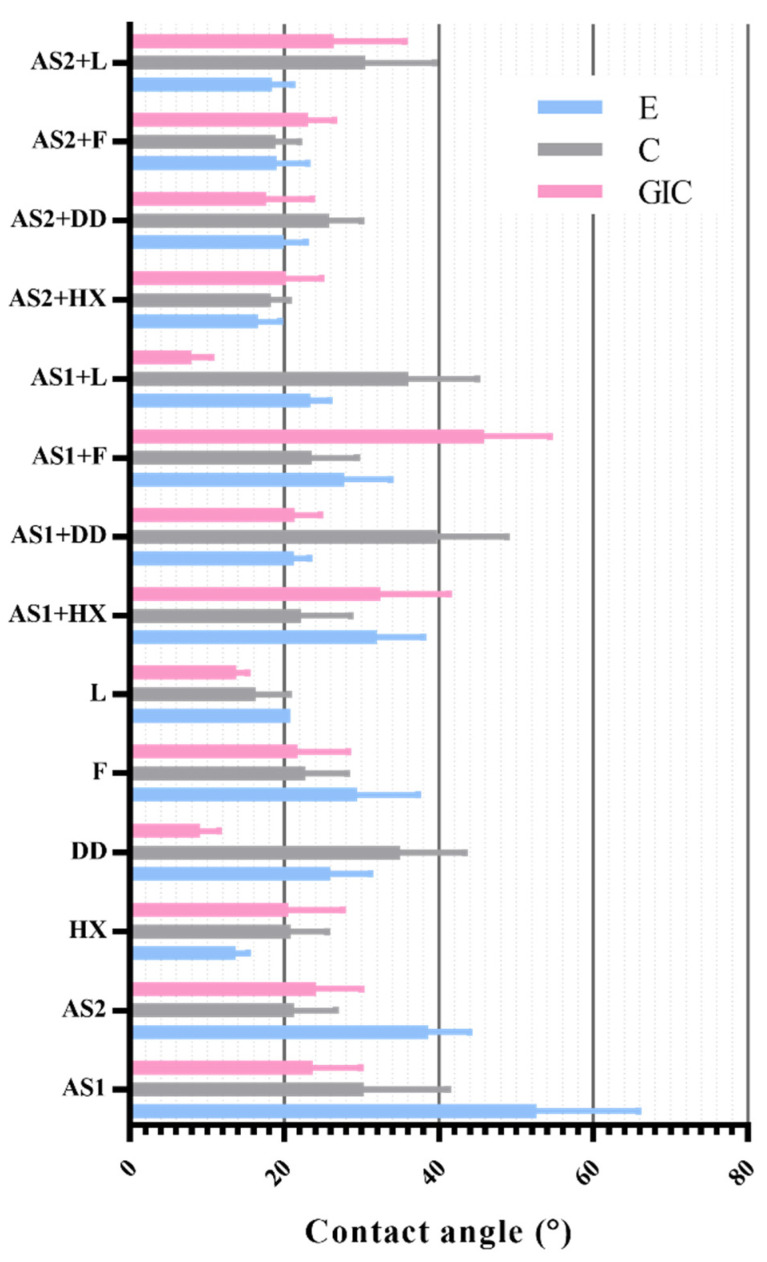
Contact angles between investigated liquids and enamel (E), glass-ionomer (GIC), and composite material (C).

**Figure 6 materials-15-04533-f006:**
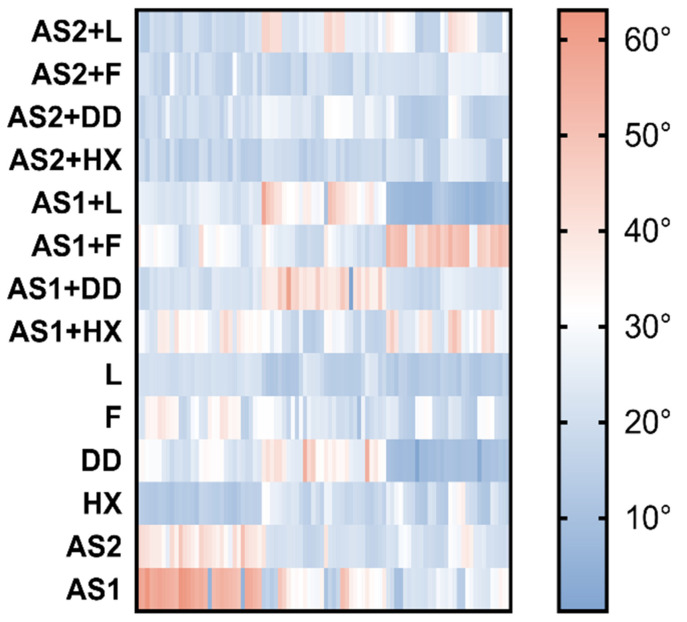
Heat map showing the contact angle range in 14 analyzed specimens in all measurements.

**Figure 7 materials-15-04533-f007:**
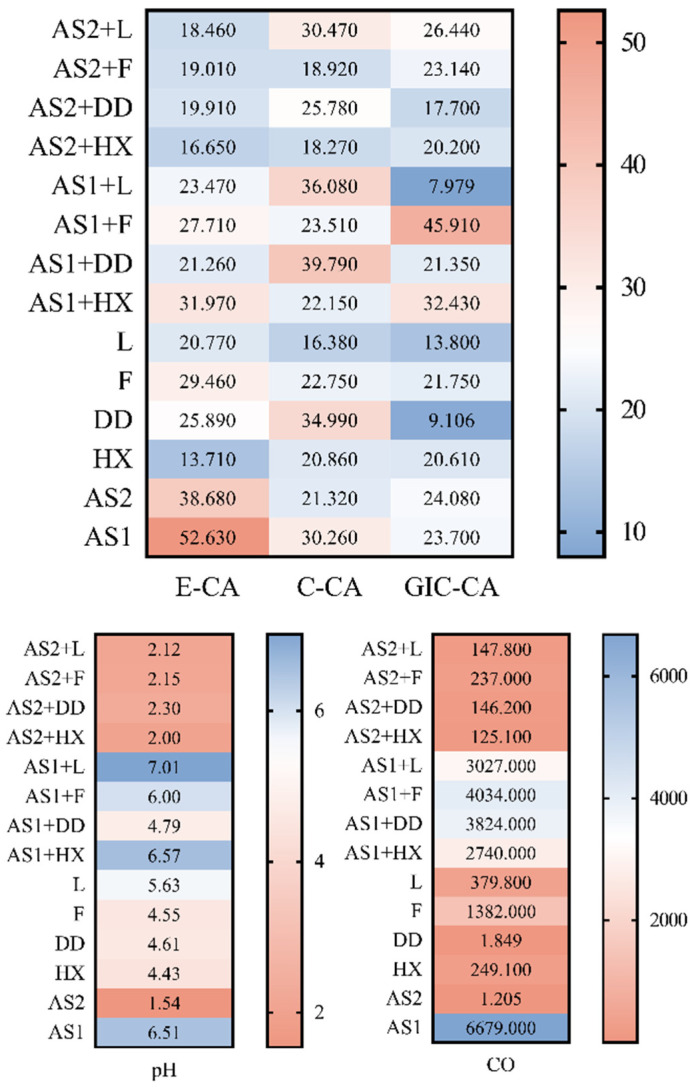
Heat map showing the relationship among contact angle range, pH range, and conductivity range in 14 analyzed specimens.

**Table 1 materials-15-04533-t001:** Mean values and standard deviation of the contact angles between investigated liquids and enamel, glass-ionomer and composite material.

	E	C	GIC
	Mean ± St. Dev.	Mean ± St. Dev.	Mean ± St. Dev.
AS1	52.63 ± 13.210	30.26 ± 10.94	23.70 ± 6.195
AS2	38.68 ± 5.327	21.32 ± 5.402	24.08 ± 5.939
HX	13.71 ± 1.622	20.86 ± 4.685	20.61 ± 6.992
DD	25.89 ± 5.279	34.99 ± 8.429	9.11 ± 2.421
F	29.46 ± 7.862	22.75 ± 5.407	21.75 ± 6.606
L	20.77 ± 1.045	16.38 ± 4.252	13.80 ± 1.475
AS1 + HX	31.97 ± 6.073	22.15 ± 6.416	32.43 ± 8.901
AS1 + DD	21.26 ± 2.019	39.79 ± 8.999	21.35 ± 3.32
AS1 + F	27.71 ± 6.108	23.51 ± 5.935	45.91 ± 8.466
AS1 + L	23.47 ± 2.433	36.08 ± 8.883	7.98 ± 2.608
AS2 + HX	16.65 ± 2.827	18.27 ± 2.412	20.20 ± 4.644
AS2 + DD	19.91 ± 2.884	25.78 ± 4.258	17.70 ± 5.941
AS2 + F	19.01 ± 4.041	18.92 ± 3.012	23.14 ± 3.371
AS2 + L	18.46 ± 2.631	30.47 ± 8.954	26.44 ± 9.153

**Table 2 materials-15-04533-t002:** Contact angles, automatic temperature compensation within the conductivity measurement canal (ATC), pH, and conductivity of the tested solutions at room temperature.

	C (°)	E (°)	GIC (°)	ATC (T °C)	T (°C)	PH	CO (μS/CM)
**AS1**	30.3	52.6	23.7	25.0	22.7	6.51	6679.0
**AS2**	21.3	38.7	24.1	26.4	25.2	1.54	1.2
**HX**	20.9	13.7	20.6	25.2	19.8	4.43	249.1
**DD**	35.0	25.9	9.11	25.2	20.0	4.61	1.8
**F**	22.7	29.5	21.8	25.8	23.1	4.55	1382.0
**L**	16.4	20.8	13.8	26.5	22.8	5.63	379.8
**AS1 + HX**	22.2	32.0	32.4	26.1	23.2	6.57	2740.0
**AS1 + DD**	39.8	21.3	21.4	26.5	24.3	4.79	3824.0
**AS1 + F**	23.5	27.7	45.9	26.0	23.8	6.00	4034.0
**AS1 + L**	36.1	23.5	7.98	25.5	24.1	7.01	3027.0
**AS2 + HX**	18.3	16.6	20.2	25.5	23.8	2.00	125.1
**AS2 + DD**	25.8	19.9	17.7	25.5	25.0	2.30	146.2
**AS2 + F**	18.9	19.0	23.1	25.5	25.0	2.15	237.0
**AS2 + L**	30.5	18.5	26.4	25.2	25.0	2.12	147.8

**Table 3 materials-15-04533-t003:** Contact angles, pH, and conductivity of the tested solution at room temperature.

	Minimum			Maximum			Shapiro–Wilk *p*		
	C	E	GIC	C	E	GIC	C	E	GIC
**AS1**	10.9	5.7	9.77	51.4	63.1	35.1	0.04	<0.001	0.252
**AS2**	16.4	25.2	17.2	39.7	50.6	38.5	<0.001	0.387	0.002
**HX**	14.6	10.7	11.7	31.2	16.9	36.8	0.063	0.834	0.059
**DD**	24.2	18.2	0.77	55.7	33.6	12.5	0.017	0.002	<0.001
**F**	9.84	14.8	12.8	31.8	39.9	32.7	0.581	0.006	0.011
**L**	11.6	18.6	11.6	24.8	23.3	16.5	0.002	0.256	0.187
**AS1 + HX**	13.7	21.3	20.7	33.3	43.7	50.1	0.004	0.136	0.008
**AS1 + DD**	0.276	17.4	15.1	59.4	25.4	28.1	<0.001	0.07	0.711
**AS1 + F**	15.9	16.4	24.2	39.7	42	53.8	0.008	0.071	<0.001
**AS1 + L**	10.8	18.8	4.66	58.3	28	14	0.2	0.359	0.013
**AS2 + HX**	13	12	12.7	22.2	21.2	26.6	0.489	0.025	0.006
**AS2 + DD**	17.1	15.3	12.1	32.9	27.6	33	0.464	0.014	<0.001
**AS2 + F**	12.5	14.5	16.8	23.2	30.4	27.8	0.038	<0.001	0.042
**AS2 + L**	20.8	13.4	13.7	45.1	23.2	43.5	<0.001	0.388	0.02

**Table 4 materials-15-04533-t004:** Kruskal–Wallis analysis.

	*χ* ^2^	DF	*p*
**AS1**	48.1	2	<0.001
**AS2**	51.2	2	<0.001
**HX**	36.1	2	<0.001
**DD**	68.7	2	<0.001
**F**	16.9	2	<0.001
**L**	40.7	2	<0.001
**AS1 + HX**	26.7	2	<0.001
**AS1 + DD**	51.7	2	<0.001
**AS1 + F**	50.3	2	<0.001
**AS1 + L**	75.3	2	<0.001
**AS2 + HX**	12.2	2	0.002
**AS2 + DD**	37.5	2	<0.001
**AS2 + F**	21.2	2	<0.001
**AS2 + L**	32.7	2	<0.001

## Data Availability

Study data are available from the authors upon request; however, no identifying information on the study participants is available, in line with the applicable data protection regulation laws and ethics guidelines.

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
