# Peer review of "The Measurement of Contact Angle, pH, and Conductivity of Artificial Saliva and Mouthwashes on Enamel, Glass-Ionomer, and Composite Dental Materials"

_materials, 2022, doi:10.3390/ma15134533_

Round 1

Reviewer 1 Report

The authors overcame the problem of heterogeneity and complexity of both the substrate and the liquids in the conducted research by repeating the measurements in identical conditions. These results are interesting and suitable for a publication in Materials.

Reviewer 2 Report

Thanks for your job. I find it very interesting, but I ask for a minor revision because your bibliography seems very old to me. some items are over 30 years old. Update the bibliography.

Author Response

ANSWER TO REVIEWERS

Dear Editor and Reviewers,

Initially, we would like to thank the esteemed editor and all five reviewers for their time invested into reading our work and providing us with their constructive comments, suggesting minor revision. A point-to-point response follows below. each reviewer points are displayed in italic font while answers are given in blue colored text

Comment

Thanks for your job. I find it very interesting, but I ask for a minor revision because your bibliography seems very old to me. some items are over 30 years old. Update the bibliography..

Answer:

We would like to thank the esteemed reviewer for their time and their kind words about our work. We hope that by incorporating the changes requested by their other colleagues, the new version will be an even more streamlined representation of our work and its importance to the scientific field. We absolutely agree with the statement that some of our literature sources were very old in the previous version of the manuscript. In the revised manuscript the following references were omitted:

  1. Edgar, W. M. Saliva: Its secretion, composition and functions. Br. Dent. J. 1992,172(8), 305–312. https://doi.org/10.1038/sj.bdj.4807861
  2. Kwok, D. Y., Neumann, A. W. Contact angle measurement and contact angle interpretation. Adv. Colloid Interface Sci. 1999,81,3, 167–249. https://doi.org/10.1016/S0001-8 686(98)00087-6
  3. Vissink, A.; ’s-Gravenmade, E.J.; Panders, A.K.; Vermey, A.; Petersen, J.K.; Visch, L.L.; Schaub, R.M.H. A Clinical Comparison between Commercially Available Mucin- and CMC-Containing Saliva Substitutes. International Journal of Oral Surgery 1983, 12, 232–238, doi:10.1016/S0300-9785(83)80048-9.
  4. Amirfazli, A.; Kwok, D.Y.; Gaydos, J.; Neumann, A.W. Line Tension Measurements through Drop Size Dependence of Contact Angle. Journal of Colloid and Interface Science 1998, 205, 1–11, doi:10.1006/jcis.1998.5562.
  5. Gaydos, J.; Neumann, A.W. The Dependence of Contact Angles on Drop Size and Line Tension. Journal of Colloid and Interface Science 1987, 120, 76–86, doi:10.1016/0021-9797(87)90324-9.
  6. Vissink, A.; Waterman, H.A.; ’s-Gravenmade, E.J.; Panders, A.K.; Vermey, A. Rheological Properties of Saliva Substitutes Containing Mucin, Carboxymethylcellulose or Polyethylenoxide. J Oral Pathol Med 1984, 13, 22–28, doi:10.1111/j.1600-0714.1984.tb01397.x.
  7. Bonesvoll, P.; Lökken, P.; Rölla, G.; Paus, P.N. Retention of Chlorhexidine in the Human Oral Cavity after Mouth Rinses. Archives of Oral Biology 1974, 19, 209–212, doi:10.1016/0003-9969(74)90263-5.
  8. Cummins, D.; Creeth, J.E. Delivery of Antiplaque Agents from Dentifrices, Gels, and Mouthwashes. J Dent Res 1992, 71, 1439–1449, doi:10.1177/00220345920710071601.
  9. Perdok, J.F.; van der Mei, H.C.; Busscher, H.J. Physicochemical Properties of Commercially Available Mouthrinses. Journal of Dentistry 1990, 18, 147–150, doi:10.1016/0300-5712(90)90055-J.
  10. Perdok, J.F.; Busscher, H.J.; Weerkamp, A.H.; Arends, J. The Effect of an Aminefluoride-Stannous Fluoride-Containing Mouthrinse on Enamel Surface Free Energy and the Development of Plaque and Gingivitis. Clin Prev Dent 1988, 10, 3–9.
  11. Edgar, W.M. Saliva: Its Secretion, Composition and Functions. Br Dent J 1992, 172, 305–312, doi:10.1038/sj.bdj.4807861

and replaced with newer, pertinent references with some minor text changes, accordingly:

  1. Fatima, S.; Rehman, A.; Shah, K.; Kamran, M.; Mashal, S.; Rustam, S.; Sabir, M.; Nayab, A.; Muzammal, M. Composition and function of saliva: a review. World journal of pharmacy and pharmaceutical sciences 2020, 9, 1552–1567, doi:10.20959/wjpps20206-16334
  2. Jaiswal, N.; Patil, P.G.; Gangurde, A.; Parkhedkar, R.D. Wettability of 3 Different Artificial Saliva Substitutes on Heat-Polymerized Acrylic Resin. The Journal of Prosthetic Dentistry 2019, 121, 517–522, doi:10.1016/j.prosdent.2018.03.037.
  3. Farinone, M.; Foglio Bonda, A.; Foglio-Bonda, Pi.L.; Pattarino, F. Artificial saliva substitutes evaluation: the role of some chemical-physical properties. 2018, 7, 67–70.

21.Drelich, J.W. Contact Angles: From Past Mistakes to New Developments through Liquid-Solid Adhesion Measurements. Advances in Colloid and Interface Science 2019, 267, 1–14, doi:10.1016/j.cis.2019.02.002.

  1. Drelich, J.W.; Boinovich, L.; Chibowski, E.; Della Volpe, C.; HoÅ‚ysz, L.; Marmur, A.; Siboni, S. Contact Angles: History of over 200 Years of Open Questions. Surface Innovations 2020, 8, 3–27, doi:10.1680/jsuin.19.00007
  2. Hinic, S.; Petrovic, B.; Kojic, S.; Omerovic, N.; Jevremov, J.; Jelenciakova, N.; Stojanovic, G. Viscosity and Mixing Properties of Artificial Saliva and Four Different Mouthwashes. Biorheology 2021, 57, 87–100, doi:10.3233/BIR-201008
  3. Reda, B.; Hollemeyer, K.; Trautmann, S.; Hannig, M.; Volmer, D.A. Determination of Chlorhexidine Retention in Different Oral Sites Using Matrix-Assisted Laser Desorption/Ionization-Time of Flight Mass Spectrometry. Archives of Oral Biology 2020, 110, 104623, doi:10.1016/j.archoralbio.2019.104623
  4. Rajendiran, M.; Trivedi, H.M.; Chen, D.; Gajendrareddy, P.; Chen, L. Recent Development of Active Ingredients in Mouthwashes and Toothpastes for Periodontal Diseases. Molecules 2021, 26, 2001, doi:10.3390/molecules26072001

38.Ricci, H.; Scheffel, D.; de Souza Costa, C.; dos Santos, F.; Jafelicci Jr, M.; Hebling, J. Wettability of Chlorhexidine Treated Non-Carious and Caries-Affected Dentine. Australian Dental Journal 2014, 59, 37–42, doi:10.1111/adj.12150.

39.Hannig, C.; Gaeding, A.; Basche, S.; Richter, G.; Helbig, R.; Hannig, M. Effect of Conventional Mouthrinses on Initial Bioadhesion to Enamel and Dentin in Situ. CRE 2013, 47, 150–161, doi:10.1159/000345083.

40.Kaga, N.; Nagano-Takebe, F.; Nezu, T.; Matsuura, T.; Endo, K.; Kaga, M. Protective Effects of GIC and S-PRG Filler Restoratives on Demineralization of Bovine Enamel in Lactic Acid Solution. Materials (Basel) 2020, 13, 2140, doi:10.3390/ma13092140.

Reviewer 3 Report

-Please check the language in the last sentence in the abstract (When an experiment uses human material as substrate...........)

- Please remove (ph) from the keywords 

-Line 52 in the introduction: this paragraph is long and only the authors added two Refs, I suggest to add more references in the middle of this paragraph 

-The introduction is long, you can remove some unnecessary sentences

-please check the subscription (such as Al2O3...etc)

-please write (table) as (Table)

-can you add some equations used in this work in section 2

-some numerical results must be added to the conclusion

Author Response

ANSWER TO REVIEWERS

Dear Editor and Reviewers,

Initially, we would like to thank the esteemed editor and all five reviewers for their time invested into reading our work and providing us with their constructive comments, suggesting minor revision. A point-to-point response follows below. each reviewer points are displayed in italic font while answers are given in blue colored text

Comment: -Please check the language in the last sentence in the abstract (When an experiment uses human material as substrate...........)

Answer: We are really thankful for pointing on this, a bit vague sentence in the abstract section. This sentence was rephrased as follows:

“When an experiment uses human material and bioactive materials which are used in biomedicine as substrate, and bioactive materials used in biomedicine additional definition of protocols is highly recommended for the future research of this topic”

Comment: Please remove (ph) from the keywords

Answer: We appreciate the observation of the reviewer. In response, we have now removed “ph” from the keywords

Comment: Line 52 in the introduction: this paragraph is long and only the authors added two Refs, I suggest to add more references in the middle of this paragraph

Answer: We appreciate this comment. We agree with the observation that mentioned paragraph was too long. The introduction section was subsequently rewritten in order to include the abovementioned points. Additional references were added in the revised version of the manuscript.

Comment: The introduction is long, you can remove some unnecessary sentences

Answer: We would like to thank the reviewer for pointing this out, the section has been rewritten as per the reviewer’s suggestion. Some unnecessary sentences were removed in the revised version of the manuscript.

Comment: -please check the subscription (such as Al2O3...etc)

Answer: We would like to thank the reviewer for drawing our attention to this typing errors. Subscriptions were correctly used in the present version of the manuscript.

Comment. -please write (table) as (Table)

Answer: The reviewer’s point is valid, it was a typesetting error on our behalf, and it has now been amended. The word “table” has been written as “Table” consistently throughout the revised version of  the manuscript.

Comment: -can you add some equations used in this work in section 2

Answer: We are greatly appreciative to the reviewer for highlighting this. 5 equations were added to section 2 which were crucial in the calculation process, with an expansion to the related paragraphs. The modified paragraph is as follows:

The next two segments, line and circle extraction were done using the Hough Transform. For line extraction, equation (1) was used. After which the extracted line was drawn as a visual representation of the border between substrates.

(1)

The circle extraction was more complex, as it was designed to search for the best radius and the best coordinates using the equations (2a, 2b, 2c, 2d), and the algorithm described in [18]. This was critical in completely eliminating any user interaction with the image.

(2a)

(2b)

(2c)

(2d)

Comment: -some numerical results must be added to the conclusion.

Answer: We would like to thank the reviewer for pointing this out, the conclusion section has been rewritten as per the reviewer’s suggestion, and numerical values were added as follows:

In this paper we presented the contact angle, pH and conductivity of 2 artificial saliva, 4 mouthwashes and their mixtures. The liquids are tested on human enamel, glass-ionomer and composite. The main findings point to the fact that all tested liquids exhibited hydrophilic behavior, but this behavior significantly varied between the tested liquids, their mixtures and in regard to analyzed substrate. Extreme heterogeneity had been observed between contact angle, acidity and conductivity.  Observed pH values ranged from 1.54, in salivation stimulants to 7.01 with artificial saliva solution and chlorhexidine solution. A wide range of conductivity values were obtained from 1,205 mS/cm in saliva stimulators to 6679 mS/cm in artificial saliva. Despite the fact that obtained data corelated with literature, sometimes acidity decreased below critical levels of 5.5 pH which is clinically relevant. Each liquid exhibited specific behavior on each substrate. The used algorithm aims to remove bias and human error since it is fully automatized. Besides, used algorithm enables importing a folder of images instead inserting images one by one.

 When an experiment uses human material as substrate and bioactive materials used in biomedicine additional definition of protocols is highly recommended for the future research of this topic.

Reviewer 4 Report

In this study, the analysis of contact angle, pH, and conductivity on  enamel, glass ionomer and composite was evaluated. Some comments were given in order to improve the manuscript.

Materials and Methods

1. All the static contact angle experiments were done at room temperature, 25°C using customized setup.  Did you think about to measure the static contact angle at 37C (simulate the oral environment)? The sample temperature was regulated with a temperature controlled plate.

Results

1. Figs. 3c and 3d are not clear. Please check.

2. Please check Tables 1, 2 and 3, the alignment of table column.

3. Table 2, please explain ATCT and T. In addition, for T, there is variation from 19.8 to 25.2. Is it the change of temperature during contact angle measurement?

Discussion

1. Could you explain briefly the effect of conductivity on the contact angles? The conductivity values were measured but its effect on contact angles was not discussed.

2. Could you explain in more details how to intercept the heat maps, Figs. 6 and 7?

3.Line 415-420, the discussion of the relationship between surface wettability and bacterial adhesion. This was discussed in Introduction (line 64-69). Please check.  

4. Line 450-453, please check the solubility of CaF2 at different pH values. CaF2 is more soluble at acidic pH (< 3). A protective coating could be formed at higher pH (> 5).

Please check the English and spelling, e.g. Ph => pH. 

Author Response

ANSWER TO REVIEWERS

Dear Editor and Reviewers,

Initially, we would like to thank the esteemed editor and all five reviewers for their time invested into reading our work and providing us with their constructive comments, suggesting minor revision. A point-to-point response follows below. each reviewer points are displayed in italic font while answers are given in blue colored text

Comment: In this study, the analysis of contact angle, pH, and conductivity on  enamel, glass ionomer and composite was evaluated. Some comments were given in order to improve the manuscript.

Answer: We appreciate the recognition of the importance of our work by the reviewer. We would like to thank the reviewer for the favorable opinion and valuable suggestions for the improvement of the manuscript.

Comment. Materials and Methods

  1. All the static contact angle experiments were done at room temperature, 25°C using customized setup. Did you think about to measure the static contact angle at 37C (simulate the oral environment)? The sample temperature was regulated with a temperature controlled plate.

Answer: We appreciate the reviewer’s viewpoint. It is true that at low temperature values, contact angles show a small and linear decrease with temperature. For higher temperature values, substantially larger decreases are exhibited. Before starting the experiment we had the same concern, thinking about the possibility and applicability of conducting the experiment at simulated physiological environment temperature, 37º C, but this is our explanation why we choose to perform the analysis on room temperature:

First, as suggested by the literature data*, the temperature dependence of contact angle of water may be classified into three regimes:

(a) low temperatures below the saturation point (i.e., 100 °C at atmospheric pressure),

(b) medium temperatures up to ~170 °C, and

(c) high temperatures up to 300 °C at pressurized conditions.

*-1. Song, J.-W.; Fan, L.-W. Temperature Dependence of the Contact Angle of Water: A Review of Research Progress, Theoretical Understanding, and Implications for Boiling Heat Transfer. Advances in Colloid and Interface Science 2021, 288, 102339, doi:10.1016/j.cis.2020.102339.

Since both the room temperature and the temperature of 37 degrees are in the same,  first group, that was one of the  to conduct the whole experiment in room temperature conditions.

Additionally, we  followed „classical” recommendation where for some surfaces observed no variation of contact angle from room temperature to 60° C for water at atmospheric pressure, although coefficients of  −0.3 degree °C−1 and −0.15 degree °C−1 were reported for the some systems, complex liquids, such as butyl chloride and n-heptane and butyl alcohol. In the same article it has been also quoted contact angle invariance with temperature within the range 0° to 60° C for the ‘Teflon’–water system at a pressure of 1 atmosphere*. And because of all this, we concluded that conducting an experiment at a temperature of 37 degrees would contribute very little to the accuracy of the measurement, and would require significant methodological difficulties.

*1. Ponter, A.B.; Boyes, A.P. Temperature Dependence of Contact Angles of Water on a Low Energy Surface under Conditions of Condensation and at Reduced Pressures. Nature Physical Science 1971, 231, 152–153, doi:10.1038/physci231152a0.

Out of available and cited literature sources the following experiments had been carried out on room temperature:

  • Kugel, G.; Klettke, T.; Goldberg, J.A.; Benchimol, J.; Perry, R.D.; Sharma, S. Investigation of a New Approach to Measuring Contact Angles for Hydrophilic Impression Materials. J Prosthodont 2007, 16, 84–92, doi:10.1111/j.1532-849X.2007.00164.x.
  • Lam, C.N.C.; Kim, N.; Hui, D.; Kwok, D.Y.; Hair, M.L.; Neumann, A.W. The Effect of Liquid Properties to Contact Angle Hysteresis. Colloids and Surfaces A: Physicochemical and Engineering Aspects 2001, 189, 265–278, doi:10.1016/S0927-7757(01)00589-1.
  • Stojicic, S.; Zivkovic, S.; Qian, W.; Zhang, H.; Haapasalo, M. Tissue Dissolution by Sodium Hypochlorite: Effect of Concentration, Temperature, Agitation, and Surfactant. Journal of Endodontics 2010, 36, 1558–1562, doi:10.1016/j.joen.2010.06.021.
  • Jaiswal, N.; Patil, P.G.; Gangurde, A.; Parkhedkar, R.D. Wettability of 3 Different Artificial Saliva Substitutes on Heat-Polymerized Acrylic Resin. Journal of Prosthetic Dentistry 2019, 121, 517–522, doi:10.1016/j.prosdent.2018.03.037.
  • Milić, L., Qureshi, S., Vejin, M., Stanojević, K., Kojić, S., Petrović, B., Stojanović, G. Automation of measurement of wetting angle of human enamel with antiseptic solutions using MATLAB, 13th Scientific Conference for Metrology and Quality in Production Engineering and Environmental Protection - ETIKUM’2021, 3-4 December 2021, Novi Sad, Serbia
  • Abdelsalam, R. (n.d.). Fluid Contact Angle Assessment to Evaluate Wetting of Dental Materials [Text]. East Carolina University. Retrieved 3 December 2021, from http://libres.uncg.edu/ir/listing.aspx?id=29696
  • Claro-Pereira, D.; Sampaio-Maia, B.; Ferreira, C.; Rodrigues, A.; Melo, L.F.; Vasconcelos, M.R. In Situ Evaluation of a New Silorane-Based Composite Resin’s Bioadhesion Properties. Dent Mater 2011, 27, 1238–1245, doi:10.1016/j.dental.2011.08.401.

The following had been carried out on temperature of 37º C

  • Vissink, A.; De Jong, H.P.; Busscher, H.J.; Arends, J.; Gravenmade, E.J. Wetting Properties of Human Saliva and Saliva Substitutes. J Dent Res 1986, 65, 1121–1124, doi:10.1177/00220345860650090301.

In the following manuscripts the temperature data had not been mentioned at all.:

  • SciELO - Brazil - Surface Properties of Dental Polymers: Measurements of Contact Angles, Roughness and Fluoride Release Surface Properties of Dental Polymers: Measurements of Contact Angles, Roughness and Fluoride Release Available online: https://www.scielo.br/j/mr/a/ky9CD8VLcWYKyKQd4ChvYjq/abstract/?lang=en (accessed on 15 June 2022).
  • Viscosity and Wettability of Animal Mucin Solutions and Human Saliva - Park - 2007 - Oral Diseases - Wiley Online Library Available online: https://onlinelibrary.wiley.com/doi/full/10.1111/j.1601-0825.2006.01263.x?casa_token=Hko3VYP1cA8AAAAA%3AWmcyAV4I15KS-RZZeZ3uBj5cjDaBfj60PEs76BitAOPWThndIc2Owph8nX0i1eAwzdnosv1r7G0rRg (accessed on 15 June 2022).
  • Sang, T.; Ye, Z.; Fischer, N.G.; Skoe, E.P.; Echeverría, C.; Wu, J.; Aparicio, C. Physical-Chemical Interactions between Dental Materials Surface, Salivary Pellicle and Streptococcus Gordonii. Colloids and Surfaces B: Biointerfaces 2020, 190, 110938, doi:10.1016/j.colsurfb.2020.110938.

Finally, in one of the sources, ref (31) * this topic has been addressed as follows: “According to Karmouch et al., contact angle measurements are affected by surface temperatures that are below 5º C [19]. Bernardin et al. found that temperatures above 120º C also affected contact angle measurements [20]. Palamara et al. conducted an experiment on contact angles of CO2 brine quartz systems and found that the contact angle changes by 0.18º for every 1º  C change

in temperature [21]. Moreover, temperature in the oral cavity constantly varies from 0 to 55º C

depending on food and beverage intake [22]. Due to the fact that minor temperature changes cause insignificant contact angle changes and that the oral cavity does not have a consistent temperature, the influence of temperature on contact angle will be disregarded in this work. All materials used in this experiment will be maintained at a room temperature of 69 F for consistency”

* Abdelsalam, R. (n.d.). Fluid Contact Angle Assessment to Evaluate Wetting of Dental Materials [Text]. East Carolina University. Retrieved 3 December 2021, from http://libres.uncg.edu/ir/listing.aspx?id=29696

Having all the above in mind, we believe that conducting the experiment at room temperature does not represent a significant shortcoming of the conducted research. Moreover, it allows comparison of the obtained values with a large number of similar experimental studies conducted under the same conditions. Realizing the importance of the reviewer's comments, however, we decided to include this dilemma in the discussion and added the following paragraph to the current version of the text:

“Surface temperatures below 5º C and above 120º C have an impact on contact angle measurements, and the experiments have shown that for every 1º C change in tempera-ture, the contact angle changes by 0.18º [41] The majority of the similar studies had been conducted on room temperature. Furthermore, the temperature in the oral cavity fluctuates constantly between 0 and 55 degrees Celsius, depending on the foods and beverages. The effect of temperature on contact angle was ignored in this study because minor tempera-ture changes cause insignificant contact angle changes and the oral cavity does not have a consistent temperature. For consistency, all materials used in this experiment were kept at room temperature.”

Comment: Results

  1. Figs. 3c and 3d are not clear. Please check.

Answer: We would like to thank the reviewer for drawing our attention to this problem. Following this, we redrawn the figure completely and we have written a follow-u, detailed description of what the Figs. 3c and 3d are. We have redrawn the whole Figure 3, placed a proper legend and added following text:

The next two segments, line and circle extraction were done using the Hough Transform. For line extraction, equation (1a) was used. The transformation of the edge image using the Hough transform for the purpose of line detection is shown in Fig. 3c. For the purpose of avoiding undefined regions in the k-n parameter space, the transform was implemented in the polar coordinate ρ-ϑ system, using the equation (1b) and equations (1c, 1d) were used for calculating k and n, described in [18]. After which the extracted line was drawn as a visual representation of the border between substrates.

?=?∗?+?                                                                   (1a)

?∗???(?)+?∗???(?)=?                                                           (1b)

?= −???(?)                                                                 (1c)

?= ????(?)                                                                (1d)

The circle extraction was more complex, as it was designed to search for the best radius and the best coordinates using the equations (2a, 2b, 2c, 2d), and the algorithm described in [18]. Fig.3d shows the edge image in the p-q space, mapped using equation (2a). It is important to note that the pixel (or a group of joint pixels) with the highest intensity in p-q space represents the coordinates of the circle center in the x-y space, for a given radius. This was critical in completely eliminating any user interaction with the image.

Comment: 2 . Please check Tables 1, 2 and 3, the alignment of table column.

Answer: We are really grateful for paying attention and pointing out to this formatting error. All tables had been redrawn in the present version of the manuscript

Comment: 3. Table 2, please explain ATCT and T. In addition, for T, there is variation from 19.8 to 25.2. Is it the change of temperature during contact angle measurement?

Answer:  We are greatly appreciative to the reviewer for highlighting this. We omitted to properly designate all the data. Actually, we apologize for the typing error, the proper designation is ATC (T ºC) and it stands for Automatic Temperature Compensation, expressed in ºC. It is regular output result from the device that measured pH and conductivity. Conductivity and pH measurement were performed by a commercial pH meter and conductometer system (pH/ISE/EC Meter, Hanna Instruments, HI5222). And ATC and T are the temperatures that were measured during pH and conductivity measurements. Contact angle measurements were performed using different setup. In the revised version of the manuscript all data and tables are properly designated.

Comment. Discussion

  1. Could you explain briefly the effect of conductivity on the contact angles? The conductivity values were measured but its effect on contact angles was not discussed.

Answer: Thank you for this comment. To our knowledge, there is no literature data on the possible connection between electrical conductivity and contact angle. The aim of this study was not to determine how three important parameters, not necessarily dependent on each other, behave in the composition of saliva substitutes, mouthwashes and their mixtures. When it comes to real saliva, pH, conductivity and contact angle are physiologically balanced in order to preserve oral homeostasis. If necessary, the pH of the saliva increases thanks to the buffering capacity, the contact angle changes with the amount of protein, and the conductivity increases with the increase of electrolytes. The aim of the study was to examine the extent to which saliva substitutes and antiseptic solutions meet these basic physical defensive properties of true real saliva. And the results show that it is much more complex than it seems at first glance. To clarify this, the following text has been added:

“When it comes to real saliva, pH, conductivity and contact angle are physiologically balanced in order to preserve oral homeostasis. If necessary, the pH of the saliva increases thanks to the buffering capacity, the contact angle changes with the amount of protein, and the conductivity increases with the increase of electrolytes. The aim of the study was to examine the extent to which saliva substitutes and antiseptic solutions meet these basic physical defensive properties of real saliva”

Comment: 2. Could you explain in more details how to intercept the heat maps, Figs. 6 and 7?

Answer: We appreciate this comment. The aim of the heat map as result representation model is to give a colorimetric view of the obtained data. Especially, in the cases where there are many measurements results, which was the case in our study. Statistics is of course more precise but heat maps are more open to general scientific community and very helpful when there is a multidisciplinary team.” Fig. 6. depicts that AS1 and solutions with AS1 with its solutions have larger contact angle than AS2 ant its solutions. Fig 7 AS1 is extremely conductive in its pure form but also in solutions and corresponds mainly to alkaline pH values. The wettability of AS1 is slightly less than the wettability of AS2. Unlike AS1, AS2 has low conductivity, ie a low amount of electrolyte that correlates with acidic pH. This acidity was minimally neutralized by diluting the solution.” Text within quotation mark was added in the revised text of the manuscript.

Comment: 3.Line 415-420, the discussion of the relationship between surface wettability and bacterial adhesion. This was discussed in Introduction (line 64-69). Please check. 

Answer: The reviewer’s point is valid, the relationship between surface wettability and bacterial adhesion was discussed earlier in the manuscript. This paragraph was omitted in the revised version of the manuscript.

Comment

  1. Line 450-453, please check the solubility of CaF2 at different pH values. CaF2 is more soluble at acidic pH (< 3). A protective coating could be formed at higher pH (> 5).

Answer: We are greatly appreciative to the reviewer for highlighting this. Although we believe that the previous statement is correct from the point of view of cariology, we agree that it requires modification from the aspect of materials science, so we have slightly modified the text.

“At a low pH, fluoride ions, for example, are more stable, and a low pH may induce the synthesis of CaF, which is hypothesized to form a, because low pH (less than 5) causes CaF2 to lose its stability and dissolve slowly.. Higher pH values enable formation of CaF2 protective coating on enamel from which F- ions are slowly released [40].”

Comment: Please check the English and spelling, e.g. Ph => pH.

Answer: The revised manuscript is revised by English native speaker and corrected accordingly. The spelling has also been doublechecked

Reviewer 5 Report

In materials and methods section, the teflon mold was a cylinder with 15mm diameter and 10mm thickness. Why did you use such dimension?

Table 1. It would be better to describe mean and standard deviation in one cell as follows.

Mean (St. dev) or Mean ± St. dev

Table 2. It may be better to explan what ‘ATCT’ and ‘CO’ mean.

Table 3. What does ‘A’ mean in the second row?

Author Response

ANSWER TO REVIEWERS

Dear Editor and Reviewers,

Initially, we would like to thank the esteemed editor and all five reviewers for their time invested into reading our work and providing us with their constructive comments, suggesting minor revision. A point-to-point response follows below. each reviewer points are displayed in italic font while answers are given in blue colored text

Comment. In materials and methods section, the teflon mold was a cylinder with 15mm diameter and 10mm thickness. Why did you use such dimension?

Answer: We are really grateful for paying attention and pointing out to this methodological detail. Honestly, we use the same or similar Teflon mold for specimen preparation for more than 15 years, as cited in the Materials and Methods section (17). Discs of this size are suitable for all types of tests, mechanical characterization and microscopy. Production and polishing are easy, preparation is not demanding, there is no breakage of the material, because these dimensions provide durability and stability.

 Comment. Table 1. It would be better to describe mean and standard deviation in one cell as follows.

Mean (St. dev) or Mean ± St. Dev

Answer: We are really grateful for paying attention and pointing out to this formatting error. All tables had been redrawn in the present version of the manuscript according to the suggestions

Comment

Table 2. It may be better to explan what ‘ATCT’ and ‘CO’ mean.

Answer: We are greatly appreciative to the reviewer for highlighting this. We omitted to properly designate all the data. Actually, we apologize for the typing error, the proper designation is ATC (T ºC) and it stands for Automatic Temperature Compensation, expressed in ºC. It is regular output result from the device that measured pH and conductivity. Conductivity and pH measurement were performed by a commercial pH meter and conductometer system (pH/ISE/EC Meter, Hanna Instruments, HI5222). And ATC and T are the temperatures that were measured during pH and conductivity measurements.

Comment Table 3. What does ‘A’ mean in the second row?.

Answer: The reviewer’s point is valid, “A” in the second row is the mistake, we are thankful for paying attention and pointing to a typesetting error on our behalf, and it has now been amended.
